

# Acute effects of muscle mechanical properties after 2000-m rowing in young male rowers

Chun-Hao Chang[1], Chin-Shan Ho[1], Fang Li[2], Chao-Yuan Chen[1,3], Hung-Chih Yeh[1] and Chia-An Ho[1]

[1] Graduate Institute of Sports Science, National Taiwan Sport University, Taoyuan City, Taiwan
[2] School of Physical Education, Central China Normal University, Wuhan City, Hubei Province, China
[3] Sport Office, National Taipei University of Business, Taipei City, Taiwan

## ABSTRACT

**Background.** The mechanical properties of muscles, such as changes in muscle tone and stiffness, are related to sports performance and injuries. Rowers are at increased risk of muscle fatigue and injury during high-repetition and heavy-load cyclic muscle actions. In view of this, the aim of the present study was to investigate the acute effect on muscle tone and stiffness, as well as bilateral muscle asymmetry, in high school rowers after a 2000-meter rowing ergometer test.

**Methods.** Twelve young male rowers (age $= 17.1 \pm 0.9$ years, body weight $= 73.5 \pm 9.7$ kg) were included in the study. The data of muscle tone (frequency) and stiffness of the posterior deltoids (PD), latissimus dorsi (LD), and rectus femoris (RF) (dominant and non-dominant side) before and after a 2000-m rowing ergometer test were collected using a handheld MyotonPRO device.

**Results.** After the rowing ergometer test, the muscle tone of dominant side PD, LD, and RF were significantly increased ($p < 0.05$). On the other hand, the muscle stiffness of the non-dominant side LD and RF, as well as the dominant side PD, LD, and RF were significantly increased after the rowing ergometer test ($p < 0.05$). The muscle tone and stiffness results showed that the dominant side PD, LD, and RF were all significantly higher than the non-dominant side after the rowing ergometer test ($p < 0.05$), where bilateral PD and RF exhibits moderate asymmetry (5% < symmetry index < 10%).

**Conclusions.** After a high-intensity and high-load 2000-m rowing ergometer test, PD, LD, and RF showed increases in muscle tone and stiffness, as well as changes in the symmetry of bilateral muscle mechanical properties.

## INTRODUCTION

Rowing is a distance-based Olympic sport which is characterized by repetitive muscle cycling and high demands on physical fitness components, such as cardiorespiratory endurance, muscular fitness, and maximal strength (*Thiele et al., 2020*), on top of rowing techniques and strategy. The standard international rowing distance involves repetitive cyclic motions with different directions repeated approximately 220–240 times during the 2000 m time trial (*Shimoda et al., 2009*). Rowers need to face sternward and paddle by

Corresponding author
Chun-Hao Chang,
hao781106@ntsu.edu.tw

using the muscle power transmitted from the lower limbs to the upper limbs to drive the leverage of oars and rowlocks to move the boat forward. The rowing motion includes two phases: the drive and recovery phases. During the drive phase, rowers paddle by extending their lower limbs and torso from a catch position, with the oars in the water, and both lower limbs and the torso are flexed. After that, during the recovery phase, the rowers prepare for the next drive phase by flexing the lower limbs and torso while the oars are positioned in the air (*Oshikawa et al., 2022*).

The biomechanics of rowing is complex, with many variables affecting the speed of the boat. Both upper and lower limbs need to be moved sequentially in order to produce optimal movement of the boat (*Wilson, Gissane & McGregor, 2014*; *Gonzalez et al., 2018*). Rowing can put significant force at several pressure points of a rower's body, such as the upper limbs, back, and lower limbs, especially with repetitive motion and intensive training. As such, the combination of high force on pressure points, intensive training, and training type puts the rower at risk of injury (*Wilson, Gissane & McGregor, 2014*). *Buckeridge, Bull & McGregor (2015)* pointed out that in rowing, rowing-induced injuries, athletic performance, and technique are all interrelated and that all these are in a state of dynamic balance. Although rowing requires great physical strength and endurance, a high level of skill and technique is critical for the efficient transfer of energy through the paddling sequence (*i.e.,* the kinetic chain). It has been found that bilateral asymmetry between the limbs can significantly affect the lumbar-pelvic kinematic parameters and pelvic torsion, which will cause asymmetric torque in the lower back, which in turn affects motion further up the kinematic chain and rowing performance (*Kingma et al., 1998*; *Buckeridge, Bull & McGregor, 2015*).

In the same way, previous studies have indicated that the mechanical properties of muscles, especially muscle tone and stiffness, are related to sports performance and sports-induced injuries. *Valderrabano et al. (2007)* and *Bravo-Sánchez et al. (2019)* pointed out that habitually loaded activities induce changes in the structural and mechanical properties of athletes' tendons and muscles, especially between the dominant and non-dominant side limbs. Most sports-induced injuries are overuse injuries, mainly manifested as contractures or muscle shortening, characterized by increased muscle tension and stiffness in addition to changes in muscle elasticity (*Iwata et al., 2019*; *McGowen et al., 2023*). For example, a prospective study of lower limb stiffness and injury incidence among elite netball players found that during the season, the mean quasi-static stiffness of the soleus muscles and Achilles tendon of injured elite players was significantly higher than that of the uninjured elite players (*Pickering Rodriguez et al., 2017*). Asymmetry in upper limb muscles (biceps brachii, posterior deltoid), back muscles (*e.g.,* latissimus dorsi, erector spinae), and lower limb muscles (*e.g.,* hamstrings, quadriceps) have been demonstrated to be associated with load and injury patterns in rowers (*Penichet-Tomás & Pueo, 2017*; *Fujita et al., 2020*; *Kabaciński et al., 2020*; *Baker, Buchanan & Bemben, 2022*). In addition, the mechanical properties of muscles and tendons may affect rapid force production and the rate of force development during dynamic movements, as they affect the stretching–shortening cycles (SSC), which are characterized by pre-stretch or counter-movements (*Monte & Zignoli,*

*2021*; *Taş, Aktaş & Tüfek, 2022*). Considering these factors, the mechanical properties of muscles can provide useful insight into injury risk monitoring and prevention strategies.

In the past, the muscle mechanical properties parameters of athletes were mainly measured by EMG in laboratory settings. In this approach, the collected data need to be further analyzed through complicated estimations, so it cannot provide real-time information on the current mechanical properties of the muscles of athletes. Another non-invasive technique for assessing the mechanical properties of muscles and tendons is the myotonometry method. The myotonometry method has been widely used in athlete, *e.g.*, soccer, tennis, Karate, breakdancer, *etc.* (*Nuñez et al., 2022*; *Colomar, Corbi & Baiget, 2022*; *Pozarowszczyk et al., 2017*; *Young et al., 2018*), to assess their muscle tone and stiffness since the introduction of the MyotonPRO device (*Amirova et al., 2021*; *Cruz-Montecinos et al., 2022*).

The bilateral limb muscles, especially the shoulder muscles, back muscles, and leg muscles, are particularly important in rowing (*Thornton et al., 2017*; *Oshikawa et al., 2022*). As during the driving phase, the muscles in these areas provide the largest source of propulsion (*Miguel et al., 2021*; *Willwacher et al., 2021*; *Oshikawa et al., 2022*), the risk of muscle fatigue and injury increases during high-repetition and heavy-load muscle cycling (*Thornton et al., 2017*), even with bilateral muscle asymmetry (*Buckeridge et al., 2012*). Previous research has pointed out that bilateral asymmetry will affect rowers' kinematics and kinetics in rowing action. This phenomenon is more obvious in sweepers than in scullers (*Buckeridge, Bull & McGregor, 2015*). Even in scull rowing, the relatively symmetrical rowing action is still affected by the inboard length of the oars used during on-water sculling, the oar handles must overlap when the blades are perpendicular to the boat (*Warmenhoven et al., 2018*); In addition, in order to keep the boat in the middle of the track under the influence of waves, the arms on both sides use different muscle strength and joint angles to control the oars during the drive phases (*Thornton et al., 2017*), resulting in upper body postural asymmetry throughout the stroke cycle.

The sequence of force generation in the rowing action is a series of kinetic chain performances from the legs to the trunk, through the shoulders, and finally to the arms (*Thornton et al., 2017*). The peak neuromuscular activities of shoulders, back, and quadriceps muscles as well as peak ergometer handle forces were all found to occur in the drive phase, lending support to this suggested mechanism of injury (*Ogurkowska, Kawalek & Zygmanska, 2015*; *Thornton et al., 2017*). Moreover, according to statistics, youth rowers have a higher annual aggregate injury rate than senior rowers (*Smoljanovic et al., 2009*; *Smoljanovic et al., 2015*). With that in mind, in our study, we selected the posterior deltoids (PD), latissimus dorsi (LD), and rectus femoris (RF) as representatives of muscles with high injury rates among the three main limb segments, and further analyzed the mechanical properties of these muscles on both sides. To the best of our knowledge, this study might be the first article about the muscle mechanical properties of high school rowers assessed by myotonometry. Therefore, the aim of the present study was to investigate the acute effect on muscle tone and stiffness, as well as bilateral muscle asymmetry, in rowers after a 2000-meter rowing ergometer test. We hypothesized that the 2000-meter rowing ergometer test would cause an acute increase in muscle tone and stiffness and that rowers would exhibit

**Table 1  The characteristics and rowing performance of participants.**

|  | ($n = 12$) |
|---|---|
| Age (years) | $17.0 \pm 0.9$ |
| Height (cm) | $171.8 \pm 5.5$ |
| Body weight (kg) | $73.5 \pm 9.7$ |
| BMI (kg/m$^2$) | $24.6 \pm 3.1$ |
| Skeletal muscle mass (kg) | $34.3 \pm 3.9$ |
| Body fat Percentage (%) | $16.6 \pm 7.3$ |
| Rowing experience (years) | $5.7 \pm 0.9$ |
| Rowing time (s) | $430.9 \pm 21.3$ |

**Notes.**

Values are reported as the mean $\pm$ standard deviation.
BMI, body mass index.

asymmetry in the mechanical properties of the bilateral limbs. The mechanical properties of the muscles may help to elucidate the potential factors that cause sports injuries in rowers.

## MATERIALS & METHODS

### Study design

In this study, the hand-held myotonometry MyotonPRO was used to investigate the muscle mechanical properties of the muscles PD, LD, and RF of rowers before and after 2000-m rowing ergometer test, and even the symmetry of the muscle mechanical properties of the bilateral limbs, which could be used to assess for overuse or fatigue. This study procedure was approved by the Institutional Review Board of Fu Jen Catholic University (New Taipei City, Taiwan) (Ethical Application Ref: C110108). All participants were fully informed of the experimental procedures and voluntarily signed informed consent before the start of the study.

### Subjects

We recruited 12 male high school rowers as the participants for our current study. The inclusion criteria were as follows: (1) active membership on a rowing team, (2) regular participation in specialized training, and (3) qualification for regional or national competitions. Individuals were excluded if they had: (1) any history of upper and lower limb muscle injury or surgery within 6 months; (2) any gait abnormality; (3) current use of pain relievers, anti-inflammatory drugs, or any medication that affects muscle tone/stiffness. The body weight, skeletal muscle mass, body mass index, and percentage of body fat of all participants were measured in this study using an InBody® 570 Body Composition Analyzer (Biospace, Inc., Seoul, Korea). Subjects were asked to complete about an hour of the test in a laboratory at a room temperature of 23 °C (on average). If a subject failed to complete the test (*e.g.*, failed to finish the 2000-m rowing ergometer test), the test was terminated immediately, and the subject's personal information and data were removed from the dataset. The subjects' anthropometric data are listed in Table 1.

## Procedures
### Mechanical properties measurement

The mechanical properties of the bilateral PD, LD, and RF (Fig. 1) were measured by a MyotonPRO device (MyotonPRO; Myoton Ltd, Tallinn, Estonia) at the pre-and post-2000-m rowing ergometer test. To measure PD muscle tone and stiffness each participant was seated in a chair, and their arm was positioned on the pillow on the laps, and the participant was asked to relax the shoulder. The head of the MyotonPRO probe was placed over the two fingerbreadths caudad to the posterior margin of the acromion (*Hung et al., 2010*). To measure LD muscle tone and stiffness each participant was placed in a supine position on a standard treatment table with the scapula was palpated by the investigator so the probe of the MyotonPRO could be placed approximately five cm above the lower angle of the scapula with the shoulder flexed at 80° (*Kurashina et al., 2022*). To measure RF muscle tone and stiffness each participant was placed in a supine lying on their backs, knees extended, and hips neutral in a relaxed position. The probe of the MyotonPRO was placed on the skin in the central part of the muscle belly perpendicular to the RF, which is the measurement point at two-thirds distally between the anterior superior iliac spine and the superior pole of the patella (*Agyapong-Badu et al., 2016*). The researcher used palpation to check that each location of the target muscle was selected correctly, and the points were marked using a skin-safe pen (*Mencel et al., 2021*). The device was then lowered to the measurement position and held steady, while the device automatically performed a predetermined standard measurement procedure. The measurement was performed on the dominant limbs before the non-dominant limbs. After the mechanical properties completed pre-test measurements, they proceeded to perform a 2000-m rowing ergometer test. To ensure that immediate post-exercise effects were collected, we repeated the same procedure within 5 min for measuring the mechanical parameters of the bilateral PD, LD, and RF immediately after the rowing ergometer test. Three repeated measurements were taken for each parameter and averaged measurements were used for statistical analysis.

### 2000-m rowing ergometer test

The 2000-m rowing ergometer test were performed on a stationary (fixed-foot stretcher) rowing ergometer (model D; Concept2; Morrisville, VT, USA), with the resistant level set at 6, which was equipped with a PM5 Performance Monitor (Concept2; Morrisville, VT, USA). During the 2,000-m rowing ergometer test, participants were asked to cover a distance of 2000-m in the shortest time possible, and the completion time was recorded. This rowing test is part of the usual training preparation and participants were familiar with the procedure (*Purge et al., 2017*).

### Measures

We collected data on the mechanical properties of the muscles PD, LD, and RF using a MyotonPRO device. MyotonPRO is a wireless handheld digital palpation device with a standard three mm diameter probe placed on the skin surface, directly over the muscle. Briefly, during the measurement of the mechanical properties of muscle with the MyotonPRO device, an initial 0.18 N force was applied to the subcutaneous tissue,

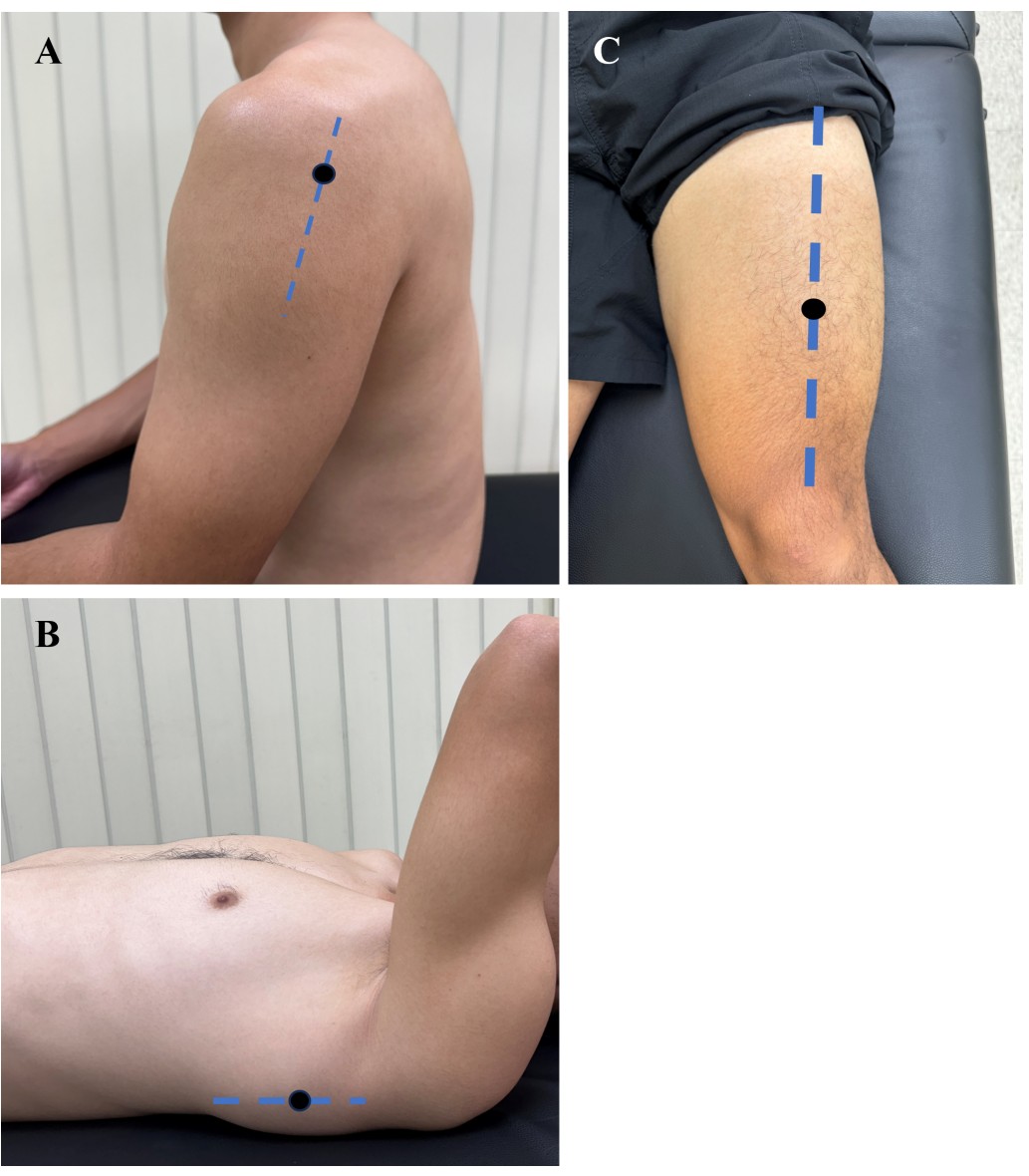

**Figure 1** **Approximate anatomic points for MyotonPro probe positioning.** (A) The posterior deltoid was measured at the 2 fingerbreadths caudad to the posterior margin of the acromion. (B) The latissimus dorsi was measured at the point approximately five cm above the lower angle of the scapula with the shoulder flexed at 80°. (C) The rectus femoris was measured at the point at two-thirds distally between the anterior superior iliac spine and the superior pole of the patella.

followed by an additional 0.4 N force for 15 ms, which resulted in muscle deformation. The resultant damped natural oscillations caused by the viscoelastic properties of soft tissues were recorded at a sampling rate of 3200 Hz through a built-in accelerometer (*Lee, Kim & Lee, 2021*). MyotonPRO has been shown to be reliable and effective for assessing the superficial muscle and tendon mechanical parameters in both clinical and laboratory settings (*Pruyn, Watsford & Murphy, 2016*; *Pozarowszczyk et al., 2017*; *Schneebeli*

*et al., 2020*; *Cruz-Montecinos et al., 2022*). This device has also been gradually used in the sports field to measure the mechanical properties of muscles and tendon in healthy and injured areas (*Nuñez et al., 2022*; *Colomar, Corbi & Baiget, 2022*), and even the acute effects after exercise or treatment (*Pozarowszczyk et al., 2017*; *Kong et al., 2018*; *Alaca & Kablan, 2022*). In this study, we analyzed the muscle tone (frequency) and stiffness parameters of the bilateral PD, LD, and RF. In this analysis, the frequency represents the natural oscillations of tissue in response to mechanical impulses. The frequency measured at rest corresponds to passive muscle tone, which is an intrinsic state of tension at the cellular level (silent EMG signal) without any voluntary contractions. Dynamic stiffness (N/m) is a mechanical property that characterizes the resistance to a contraction or to an external force that deforms its initial shape (*Schneider et al., 2015*). The formula for calculating muscle tone (frequency) (Hz) is as follows: $F = f_{max}$ (signal spectrum computed by Fast Fourier transform—FFT). The formula for calculating muscle stiffness (N/m) is as follows: $S = (\alpha_{max} \cdot m_{probe})/\Delta l$, where $\alpha$ is the acceleration of the damped oscillation, $m_{probe}$ is the mass of the measurement mechanism, and $\Delta l$ is the maximal displacement of the tissue (*MyotonPRO, 2022*).

## Data analysis

All twelve of the test properties completed the exercise tests safely, therefore all muscle tone (frequency) and stiffness data were recorded automatically by MyotonPRO software (MyotonPRO; Myoton AS, Tallinn, Estonia) and then exported in Microsoft Excel format (Excel version in Microsoft Office 2016 for Windows). In symmetry analysis, according to MyotonPRO's operation manual, the formula for calculating the symmetry index (SI, %) is defined as: $SI = (Dominant - Non \ dominant)/[(Dominant + Non \ dominant)/2] * 100$ (*MyotonPRO, 2022*). When the symmetry index of the same muscles on both sides is 5–10%, it can be regarded as moderate asymmetry; when it is greater than 10%, it can be classified as high asymmetry (*Fohanno et al., 2015*).

## Statistical analysis

The sample size was estimated with a software program (G*Power v3.1.9.7; Heinrich-Heine-Universität Dusseldorf, Dusseldorf, Germany) with an effect size of 0.5, alpha error of 0.05, and statistical power of 0.8. Thus, the minimum number of subjects required for the study was determined to be 12.

In this study, all descriptive data were summarized as means ±standard deviations. The statistical software IBM SPSS Statistics version 20 (IBM Corp., New York, NY, USA) was used for statistical analysis. We used the Kolmogorov–Smirnov test to check for normality of the data distribution. Two-way mixed-design analysis of variance (ANOVA) was used to analyze differences in muscle tone (frequency) and stiffness between dominant and non-dominant at pre- and post-rowing time-trial, where partial $\eta^2$ was calculated to assess the effect sizes. The significance level was set as $p$ less than 0.05.

## RESULTS

A total of 12 rowers completed all experimental procedures and required measurements. Table 1 shows the characteristics and rowing performance data of the participants. All

**Table 2  Comparison of the muscle tone (frequency) and stiffness for the pre and post-2000-m rowing ergometer test.**

| Session/location | Frequency (Hz) | | | Stiffness (N/m) | | |
|---|---|---|---|---|---|---|
| | Non-dominant | Dominant | *p* | Non-dominant | Dominant | *p* |
| Posterior deltoid | | | | | | |
| Pre-rowing | 12.87 ± 1.22 | 12.89 ± 1.13 | 0.93 | 164.62 ± 23.07 | 165.19 ± 22.94 | 0.91 |
| Post-rowing | 13.03 ± 1.01 | 13.84 ± 1.02* | 0.00 | 174.66 ± 25.85 | 189.61 ± 22.90* | 0.00 |
| Latissimus dorsi | | | | | | |
| Pre-rowing | 13.14 ± 1.03 | 13.15 ± 1.12 | 0.95 | 192.61 ± 21.43 | 201.54 ± 17.29 | 0.03 |
| Post-rowing | 13.78 ± 0.87 | 14.30 ± 1.26* | 0.04 | 209.28 ± 17.70* | 218.84 ± 22.45* | 0.02 |
| Rectus femoris | | | | | | |
| Pre-rowing | 15.99 ± 1.27 | 15.95 ± 1.32 | 0.92 | 275.14 ± 21.41 | 278.14 ± 29.93 | 0.66 |
| Post-rowing | 16.74 ± 1.23 | 17.60 ± 1.34* | 0.04 | 294.50 ± 22.78* | 311.64 ± 27.47* | 0.02 |

**Notes.**
Values are shown as the mean ± SD.
*Significantly different from pre-rowing, $p < 0.05$.

continuous data showed normal distributions ($p > 0.05$) by Kolmogorov–Smirnov test, meeting the statistical assumptions of parametric tests.

The comparison of the muscle tone (frequency) and stiffness for the pre and post-2000-m rowing ergometer test results are summarized in Table 2. Mixed design two-way ANOVA results showed that the frequency dominant side PD ($F = 4.62$; $p = 0.04$; $\eta^2 = 0.17$), LD ($F = 5.53$; $p = 0.03$; $\eta^2 = 0.20$), and RF ($F = 9.23$; $p = 0.01$; $\eta^2 = 0.30$) were significantly increased after the 2000-m rowing ergometer test. On the other hand, the muscle stiffness of the non-dominant side LD ($F = 4.31$; $p = 0.05$; $\eta^2 = 0.16$) and RF ($F = 4.60$; $p = 0.04$; $\eta^2 = 0.17$), as well as the dominant side PD ($F = 6.94$; $p = 0.02$; $\eta^2 = 0.24$), LD ($F = 4.48$; $p = 0.05$; $\eta^2 = 0.17$), and RF ($F = 8.16$; $p = 0.01$; $\eta^2 = 0.27$) were significantly increased after the 2000-m rowing ergometer test.

Differences in the muscle tone (frequency) and stiffness between the non-dominant side and dominant side PD, LD, and RF at the pre-and post-rowing were presented in Fig. 2. The muscle tone results of this study showed that the dominant side PD, LD, and RF were all significantly higher than the non-dominant side after the 2000-m rowing ergometer test (all $p < 0.05$). The muscle stiffness results of this study showed that the dominant side LD was significantly higher than the non-dominant side before the 2000-m rowing ergometer test ($p < 0.05$), as well as the dominant side PD, LD, and RF were all significantly higher than the non-dominant side after the 2000-m rowing ergometer test (all $p < 0.05$).

The symmetry index of the muscle tone and stiffness between the non-dominant side and dominant side at the test pre- and post-rowing were presented in Fig. 3. The muscle tone results of this study showed that before rowing, there was no significant asymmetry in the muscle tone of rowers' bilateral PD (0.16%; $p = 0.93$), LD (0.08%; $p = 0.95$), and RF (−0.25%; $p = 0.92$). After rowing, the muscle tone of rowers' bilateral PD (6.03%; $p = 0.00$), LD (3.70%; $p = 0.04$), and RF (5.01%; $p = 0.04$) reached a significant asymmetry, where PD and RF exhibits moderate asymmetry. The muscle stiffness results of this study showed that before rowing, except for rowers' bilateral LD (4.53%; $p = 0.03$), there was no significant asymmetry in the muscle tone of rowers' bilateral PD (0.35%; $p = 0.91$) and RF

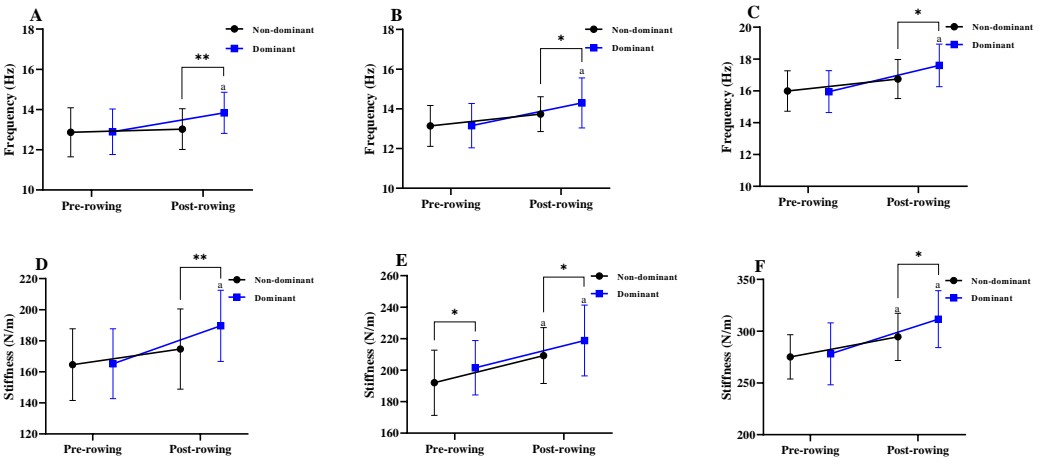

**Figure 2** **Differences in the muscle tone (frequency) and stiffness of the non-dominant side and dominant side.** Differences in the muscle tone of the posterior deltoid (A), latissimus dorsi (B), and rectus femoris (C) between the non-dominant side and dominant side. Differences in the muscle stiffness of the posterior deltoid (D), latissimus dorsi (E), and rectus femoris (F) between the non-dominant side and dominant side. Asterisks (*/**) Indicate that the dominant side was significantly different from the non-dominant side, which represents $p < 0.05$ and $< 0.01$, respectively. [a] Significantly different from pre-rowing, $p < 0.05$.

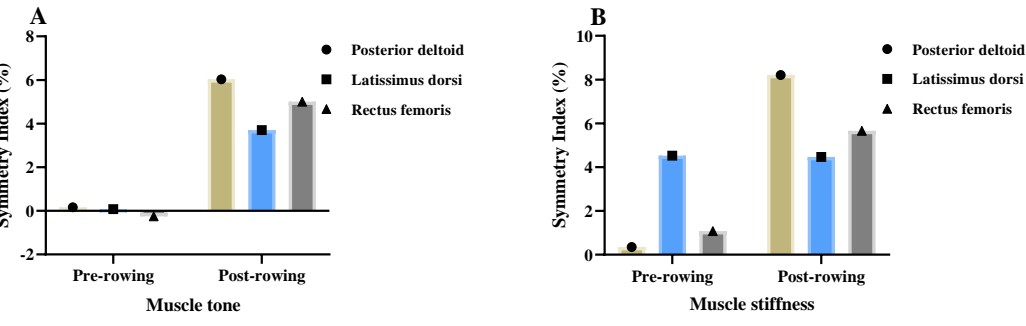

**Figure 3** **Symmetry index (%) of the muscle tone (A) and stiffness (B) between the non-dominant side and dominant side at the test pre- and post-rowing.**

(1.08%; $p = 0.66$). After rowing, the muscle stiffness of rowers' bilateral PD (8.21%; $p = 0.00$), LD (4.47%; $p = 0.02$), and RF (5.66%; $p = 0.02$) reached a significant asymmetry, where PD and RF exhibit moderate asymmetry.

## DISCUSSION

This study aims to measure the immediate effect on the mechanical properties of the bilateral PD, LD, and RF in young rowers after a 2000-m rowing ergometer test using a myotonometer, and to compare the existence of asymmetry between the dominant and non-dominant side limbs. The results showed that after the rowing ergometer test, in terms of muscle stiffness, the mean stiffness of the LD, and RF were increased on both sides, and
PD increased only on the dominant side. Regarding muscle tone, the mean frequencies of PD, LD, and RF only increase on the dominant side. Further analysis of the asymmetry of the mechanical properties of the bilateral PD, LD, and RF showed that the dominant side's mean frequency and mean stiffness were higher than those of the non-dominant side. This result illustrates the acute effect on the mechanical properties of the rowers' bilateral limbs after the rowing test, and the asymmetry of the bilateral limbs was also demonstrated in this study.

Rowing, which involves repetitive muscular motion, is a highly physiologically demanding competitive sport. Each rowing cycle is characterized by a stretching–shortening cycle (SSC) at the muscle–tendon unit (*Held et al., 2022*). Muscle tone and stiffness have been shown to affect the effectiveness of SSCs (*Ishikawa et al., 2006*; *De Ste Croix et al., 2021*). Our study showed an acute increase in muscle tone and stiffness of the PD, LD, and RF after a 2000-m rowing ergometer test, which may have been due to the tensile strain cycles of the muscle (*Shimoda et al., 2009*). *Joseph et al. (2016)* indicated that repetitive cyclic loading may affect the performance of muscle–tendon units, likely by altering muscle compliance, and these changes may lead to increased mechanical properties of muscles and tendons.

A brief mechanical impulse is sent to the muscle tissues by the handheld MyotonPRO device, and the mechanical response of the muscles is recorded through the acceleration probe. It has been proved that the mechanical properties of the muscles and tendon are thus measured with high reliability and validity (*Aird, Samuel & Stokes, 2012*; *Dellalana et al., 2019*; *Cruz-Montecinos et al., 2022*). The increase in oscillation frequency obtained from these mechanical response calculations is indicated an increase in muscle tone. The abnormally elevated muscle tone and associated intramuscular pressure may lead to impaired circulation because blood vessels in the muscle are constricted, restricting blood flow. Impaired blood circulation to the muscles can cause faster muscle fatigue during exercise and inhibit muscle recovery (*Mustalampi et al., 2012*; *Alaca & Kablan, 2022*). Increased muscle stiffness has been associated with pathological conditions such as myofascial pain and higher risk of muscle injury (*Koch & Wilke, 2022*). From our research, it can be observed that the muscle tone and stiffness of both dominant and non-dominant limbs after the rowing ergometer test were affected to varying degrees. The increase in the mechanical properties of the dominant side is more than that of the non-dominant side, which indicates the presence of inter-limb asymmetries. *Pozarowszczyk et al. (2017)* showed that the mechanical properties of Achilles tendon and soleus muscles of both dominant and non-dominant legs changed to varying degrees after completing eight kumite (sparring bouts with an opponent).

Inter-limb asymmetry has been identified as a potential factor that could lead to impaired sports performance (*Bishop, Turner & Read, 2018*) and may increase the risk of sports-induced injury (*Croisier et al., 2008*; *Mokha, Sprague & Gatens, 2016*). Long-term muscle asymmetry will put a greater burden on one side of the human body, leading to many chronic injuries and asymmetric pathological development (*Croisier et al., 2002*; *Ferber et al., 2003*; *Perttunen et al., 2004*; *Rumpf et al., 2014*). In sweep rowing, the sweeper's long-term unilateral rowing action causes asymmetry in the kinematics and kinematics

parameters between the bilateral limbs (*Buckeridge, Bull & McGregor, 2015*). However, in scull rowing, the seemingly symmetrical rowing action is still affected by factors such as the boat structure, the length of the oars, and maintaining the boat in the middle of the track. The arms must use different muscle strengths and joint angles to control the oars, resulting in the upper body postural asymmetry throughout the stroke cycle (*Thornton et al., 2017*; *Warmenhoven et al., 2018*). *Fohanno et al. (2015)* indicated that asymmetry in body movements and foot force production is detrimental to both performance and injury risk in rowers, especially when regularly practicing rowing with ergometers. According to statistics, youth rowers are more likely to be injured than adult rowers due to lack of experience, improper training, or the adaptation period when changing the sweep or scull boat (*Smoljanovic et al., 2009*; *Smoljanovic et al., 2015*).

In our study, it was found that the muscle tone and stiffness of the bilateral PD and RF between their limbs had no difference before the start of the exercise intervention, which means that when uninjured muscles are in the resting state, the mechanical properties of the muscles between the limbs are in a relatively relaxed state. However, with the intervention of the 2000-m rowing ergometer test, the muscle tone and stiffness of the bilateral PD and RF developed moderate asymmetry, with the mechanical properties of dominant side limbs higher than those of the non-dominant limbs. As with previous rowing asymmetry studies, it was confirmed after rowing ergometer testing that the rower's dominant side or limb that primarily holds the oar produced greater kinematic and kinetic differences compared to the contralateral side (*Buckeridge, Bull & McGregor, 2015*; *Fohanno et al., 2015*; *Warmenhoven et al., 2018*).

Previous studies have shown that myotonometric measurements can be used to detect relevant muscle pathomechanics in athletes (*Stefaniak, Marusiak & Bączkowicz, 2022*) and identify them as risk factors for injury and diagnostic markers of injured tissue (*Zügel et al., 2018*; *Kawai, Takamoto & Bito, 2021*). Through periodic testing, it can also help sports trainers evaluate training procedures and treatment effects (*Chang et al., 2021*). Any muscle–tendon tissue response can be affected in terms of training effectiveness and sports performance. This study found that there were differences in the mechanical properties of the bilateral PD, LD, and RF in rowers after repetitive rowing tests, which should serve as a reminder to coaches, strength trainers and athletes to adjust for asymmetric limb segments.

This study describes the acute effects of a 2000-m rowing ergometer test on the mechanical properties of the bilateral PD, LD, and RF muscles in young rowers, and the resultant observation of bilateral limb segment asymmetry. However, there are several limitations of this study. First, a greater sample size is required for a more general interpretation of these results. Second, the participants in this study were high school-aged male rowers, and the results observed may not be extrapolated to other age groups and genders, such as adult male rowers or female rowers. For future work, we will focus on the changes in mechanical properties of muscles caused by periodic training, as understanding training-induced muscle adaptations may help in the design of training programs and sports injury prevention.

## CONCLUSIONS

Our study showed that a 2000-m rowing ergometer test increased the muscle tone and stiffness of the PD, LD, and RF. The increase in muscle tone and stiffness was asymmetrically greater in the dominant limb than in the non-dominant limb, which showed moderate asymmetry in PD and RF, with a symmetry index greater than 5%. These illustrated the importance of monitoring the muscle-related mechanical properties of athletes during or immediately after training to prevent possible negative changes caused by the long-term load accumulation of daily physical training on the muscles.

## ACKNOWLEDGEMENTS

The authors acknowledge the participating rowers, and coaches for their valuable contribution to the study.

### Funding

The work was supported by the National Science and Technology Council, Taiwan (grant numbers NSTC 111-2410-H-179-002). The funders had no role in study design, data collection and analysis, decision to publish, or preparation of the manuscript.

### Grant Disclosures

The following grant information was disclosed by the authors:
National Science and Technology Council, Taiwan: NSTC 111-2410-H-179-002.

### Competing Interests

The authors declare there are no competing interests.

### Author Contributions

- Chun-Hao Chang conceived and designed the experiments, performed the experiments, analyzed the data, prepared figures and/or tables, and approved the final draft.
- Chin-Shan Ho conceived and designed the experiments, authored or reviewed drafts of the article, and approved the final draft.
- Fang Li conceived and designed the experiments, analyzed the data, prepared figures and/or tables, and approved the final draft.
- Chao-Yuan Chen performed the experiments, authored or reviewed drafts of the article, and approved the final draft.
- Hung-Chih Yeh performed the experiments, prepared figures and/or tables, and approved the final draft.
- Chia-An Ho performed the experiments, prepared figures and/or tables, and approved the final draft.

### Human Ethics

The following information was supplied relating to ethical approvals (*i.e.,* approving body and any reference numbers):

The Institutional Review Board of Fu Jen Catholic University (New Taipei City, Taiwan) (Ethical Application Ref: C110108).

## Data Availability
The raw measurements are available in the Supplementary File.

## Supplemental Information
Supplemental information for this article can be found online at http://dx.doi.org/10.7717/peerj.16737#supplemental-information.

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
