# Peer review of "Acute effects of muscle mechanical properties after 2000-m rowing in young male rowers"

_PeerJ, doi:10.7717/peerj.16737_

## Round 0.1 · original submission · Major Revisions

Although this is an interesting study for the researchers and sports community, some issues should be taken into consideration by the authors. The reviewers provided important information that should be addressed and/or clarified.

Reviewer 1 ·

Basic reporting

Thank you for the opportunity to review this manuscript.
The manuscript is clearly written, and address important issues in sport injury.

Experimental design

To enhance repeatability of the study, I suggest to include figures on the placement of MyotonPRO probe on each muscle (Line 170-180).
What is the type of the ergometer (i.e., fixed or dynamic)? Is it for sculling or sweeping? Please describe clearly.

Validity of the findings

Based on your results, please discuss thoroughly why asymmetry of muscle tone and compression stiffness were observed in a symmetry motion (ie., sculling).

Reviewer 2 ·

Basic reporting

This is an interesting study that focuses on the determination of mechanical properties of the muscles. It presents an interesting topic regarding the effects of rowing in the mechanical properties.
After analyzing the paper after the revision process, I feel the authors must address some issues and shortcoming in a major revision, before I could support its publication in this journal.

Experimental design

Abstract:
Reviewer´s comment:
Line 21: …” muscle cycling.” Please improve this sentence.
Line 21 to 24: Improve the aim of the study.

Introduction:
Reviewer´s comment:
The text should be justified in this section.
Line 51. Complete the sentence-“…exceed 220 cycles”...
Line 68-71- Please insert some references, that can support this sentence.
Line 98-99- witch where the main findings of the studies made in sport, in which sports?
Please improve these sentences to clarify the state of art with this method in sport science.
Line 103-108- the authors need to address with the support of some studies the importance of understand the muscle asymmetries, presenting some findings in this topic regarding rowing or in other sports.
Line 106- 108- Please make the state of art of mechanical properties studies made in rowing, and explain better why did the authors choose these muscles in this research?
Line 108-111- please improve the aim of this study, I suggest presenting a research question clearer with the variables of this study included.


Methods:

Subjects
Line 134-137- please improve this sentence.

Measures
The authors should use abbreviation in all the sections of the muscles studied.
Line 144 “…mechanical properties…” please change for “…mechanical properties of the muscles…”;
Line 144-151- please insert more studies that support the validation process of this device, and witch studies used this device, and witch findings they achieved.
- Please also include the explanation of the procedures for the application of this device in each athlete.
Line 152- which were the software used to treat the data, please explain each step of the data treatment in the software’s.
Line 161 and Line 162- please improve the presentation of the formulas in the text.
Line 165- Procedures, in my opinion the procedures should be presented first than the measures.
Please change the order in the text.
Line 185- 186- please indicate what was the speed of the trial? How was controlled? what did the time spend in the test for each subject, how many time they spend making the trial?

Line 193- Please insert the formulas according to the journal rules.

Validity of the findings

Results:
Table 2- please reduce the decimal cases for 2.

Discussion
Line 248- 249- please explain the main findings and if the hypothesis were confirmed or not.
Line 289-290. Please explain more deeply what you mean with “balance state”.
Line 298-301- Please remove this sentence.
Line 302- 310- The authors should be more specific regarding the practical applications to coaches and athletes, to prevent future injuries.
Authors should discuss more deeply each main finding and compare with the literature in rowing.

Conclusions:
Reviewer´s comment: the conclusion is very long, please making it shorter.

References:
Reviewer´s comment:
Please improve all the references regarding the journal instructions.
Some of the references don´t have number, or pages associated, please improve it all.

Reviewer 3 ·

Basic reporting

The authors mentioned the mechanical properties of muscles are related to performance and injury, and used the muscle tone and muscle stiffness as the outcomes. However, no relations between performance or injury were reported. It is weak to support the authors' description. The authors should state details of the importance and application of this study and explain how to use the muscle tone and muscle stiffness as indicators for sport performance and injury.

Experimental design

1. Need more subjects.
2. Why choice posterior deltoids, latissimus dorsi, and rectus femoris represented upper limb, back and lower limb.

Validity of the findings

no comment

---

## Round 0.2 · accepted · Accept

The authors have addressed all the reviewers' comments and, based on their opinion, the manuscript is ready for publication.

Reviewer 1 ·

Basic reporting

The manuscript has been improved thoroughly by the authors. I recommend to accept this manuscript.

Experimental design

No further comments.

Validity of the findings

No further comments.

Reviewer 2 ·

Basic reporting

The English grammar were improved, the references added properly.

Experimental design

Improved

Validity of the findings

Improved